# A ZTF-7/RPS-2 complex mediates the cold-warm response in *C. elegans*

**Ting Xu[1], Shimiao Liao[1], Meng Huang[1], Chengming Zhu[1], Xiaona Huang[1], Qile Jin[1], Demin Xu[1], Chuanhai Fu[1], Xiangyang Chen[1]\*, Xuezhu Feng[1]\*, Shouhong Guang[1,2]\***

**1** Department of Obstetrics and Gynecology, The First Affiliated Hospital of USTC, The USTC RNA Institute, Ministry of Education Key Laboratory for Membraneless Organelles & Cellular Dynamics, School of Life Sciences, Division of Life Sciences and Medicine, Biomedical Sciences and Health Laboratory of Anhui Province, University of Science and Technology of China, Hefei, Anhui, China, **2** CAS Center for Excellence in Molecular Cell Science, Chinese Academy of Sciences, Hefei, Anhui, China

\* xychen91@ustc.edu.cn (XC); fengxz@ustc.edu.cn (XF); sguang@ustc.edu.cn (SG)

## Abstract

Temperature greatly affects numerous biological processes in all organisms. How multicellular organisms respond to and are impacted by hypothermic stress remains elusive. Here, we found that cold-warm stimuli induced depletion of the RNA exosome complex in the nucleoli but enriched it in the nucleoplasm. To further understand the function and mechanism of cold-warm stimuli, we conducted forward genetic screening and identified ZTF-7, which is required for RNA exosome depletion from nucleoli upon transient cold-warm exposure in *C. elegans*. ZTF-7 is a putative ortholog of human ZNF277 that may contribute to language impairments. Immunoprecipitation followed by mass spectrometry (IP-MS) found that ZTF-7 interacted with RPS-2, which is a ribosomal protein of the small subunit and participates in pre-rRNA processing. A partial depletion of RPS-2 and other proteins of the small ribosomal subunit blocked the cold-warm stimuli-induced reduction of exosome subunits from the nucleoli. These results established a novel mechanism by which *C. elegans* responds to environmental cold-warm exposure.

## Author summary

Temperature is a very common stimuli for all organisms and widely affects biological processes. Physiological responses to sustain temperature changes constitute a critical adaptive mechanism for living organisms. Warm-blooded animals can maintain a fairly stable body temperature, but cold-blooded animals experience drastic shifts in body temperature. Many studies are focused on heat shock response, but how organisms respond to cold shock stimuli is largely unclear. Here, we observed that the RNA exosome, a highly conserved RNA processing machinery, responds to the hypothermic stress by altering its subcellular localization in *C. elegans*. The RNA exosome complex localized in nucleoli when the worms were grown at 20˚C. A cold-warm treatment at 4˚C for 3 hours followed by recovery at 20˚C could deplete the RNA exosome from the nucleoli to nucleoplasm. A conserved zinc finger protein ZTF-7 is required for the cold-warm shock-induced exosome translocation. ZTF-7 is an ortholog of human ZNF277 that is likely involved in

**Data Availability Statement:** The authors confirm that all data underlying the findings are fully available without restriction. The underlying numerical data for all of our graphs and summary statistics are available on figshare (https://doi.org/10.6084/m9.figshare.21972518.v1). All other data

are within the manuscript and its supplementary information.

**Funding:** This work was supported by grants from the National Key R&D Program of China (2022YFA1302700, 2019YFA0802600) to (SG), the National Natural Science Foundation of China (32230016, 91940303, 32270583, 31870812, 32070619, 31871300 and 31900434) to (SG), and the Strategic Priority Research Program of the Chinese Academy of Sciences (XDB39010600) to (SG). This study was supported in part by the Fundamental Research Funds for the Central Universities to (SG). The funders had no role in study design, data collection and analysis, decision to publish, or preparation of the manuscript.

**Competing interests:** All the authors declare no competing interest.

language impairment. ZTF-7 interacts with a small ribosomal subunit protein RPS-2. Additionally, other small ribosomal subunit proteins, for example RPS-0 and RPS-12, were also involved in the cold-warm shock-induced exosome translocation. We concluded that the ZTF-7/RPS-2 complex mediates a new cold-warm response pathway in *C. elegans*.

## Introduction

Environmental stimuli pervasively affect biological processes in all organisms. Temperature is one of the most important environmental factors that induce molecular and biochemical changes in cells. Temperature affects many physiological processes, cellular structures, and metabolic activities, as well as development and growth [1]. The elucidation of the temperature tolerance mechanism of animals is important for understanding the adaptive mechanism during evolution. Numerous investigations have focused on the characterization of heat shock-responsive genes and the mechanism by which heat shock factors (HSFs) direct the expression of heat shock proteins (HSPs). The genome-wide transcriptional response to heat shock in mammals is rapid and dynamic and results in the induction and repression of several thousand genes [2]. Heat shock transcription factors (HSFs) act as direct transcriptional activators to modulate the expression of genes that encode heat shock proteins [3].

In contrast to the heat shock response, how organisms respond to cold shock stimuli is largely unclear. In plants, COLD1 is required for chilling (0–15°C) tolerance in the Nipponbare subspecies of rice [4]. Transient receptor potential (TRP) channels are well-known temperature receptors in many species [5]. In addition, a TRP-independent temperature-sensing mechanism is also present in nematodes and flies [6,7]. In *C. elegans*, cold tolerance is regulated through ASJ sensory neuron, intestine, and sperm tissue networks [8,9]. ASJ neurons release insulin to intestinal cells to control the cold-tolerance status [8]. ADL sensory neurons were also shown to regulate cold tolerance through TRP channels [10]. GLR-3 senses cold in the sensory neuron ASER to trigger cold-avoidance behavior [11]. In addition, *C. elegans* acclimates to environmental temperature changes by changing lipid composition to regulate metabolism and behavior [8,12–15]. A genetic pathway consisting of *isy-1* and *zip-10* reacts to cold-warm stimuli and directs transcriptional responses in *C. elegans* [16].

The nucleolus has long been regarded as an rRNA- and ribosome-producing factory and plays key roles in monitoring and responding to cellular stresses [17]. The expression of rRNAs and small and large ribosomal subunits must be tightly coordinated. Meanwhile, ribosomal proteins must be precisely incorporated into dynamically folded pre-rRNAs to achieve orchestrated cell growth and proliferation [18]. Therefore, the nucleolus acts as a highly flexible and complex system responding to numerous environmental stresses. Impaired rRNA transcription and processing and nucleolar stress-induced excessive lipid accumulation could affect the survival and lifespan of *C. elegans* [19].

The eukaryotic RNA exosome is an evolutionarily conserved ribonucleolytic complex that consists of 10 or 11 subunits [20], which play important roles in RNA processing and degradation. The RNA exosome possesses a catalytically inactive barrel structure of 9-core subunits (EXO-9), arranged as a hexamer (the PH-like ring) capped with a trimeric S1/KH ring [21]. EXO-9 interacts with two 3'→5' exoribonucleases: EXOSC10 (Rrp6 in budding yeast) and DIS3 (also known as Rrp44) [22]. In humans, EXOSC10 is enriched within the nucleolus. However, canonical DIS3 is predominantly localized within the nucleoplasm [23]. In budding yeast, DIS3 is present in both nuclear and cytoplasmic exosome complexes, yet Rrp6 is found

only in the nuclear exosome complex [24]. In *C. elegans*, EXOS-10 and DIS-3 are ubiquitously expressed in all of the cells and enriched in the nucleus. Interestingly, EXOS-10 is enriched in nucleoli, while DIS-3 is enriched in the nucleoplasm but depleted in nucleoli [25]. The RNA exosome functions both in the nucleus and cytoplasm to process noncoding RNAs (ncRNAs) and degrade improperly processed (faulty) RNAs in RNA surveillance pathways [20,26–29].

Previously, we found that knocking down *M28.5*, *nol-56*, *fib-1* and *mtr-4*, the genes that are required for rRNA processing, by RNAi induced a dramatic depletion of EXOS-10 from the nucleoli [25]. Meanwhile, after knocking down these genes by RNAi, a class of antisense ribosomal siRNAs (risiRNAs) were enriched, suggesting that the proper nucleolar localization of exosomes may be important for the suppression of risiRNA production. In addition, we also found that environmental stimuli, such as low temperature, induced the accumulation of risiR-NAs that subsequently guide the nuclear Argonaute proteins NRDE-3 and NRDE-2 to the nucleoli, which are associated with pre-rRNAs and silence RNA polymerase I transcription [25,30]. Whether and how environmental stimuli could induce the translocation of exosome proteins in distinct subcellular compartments is intriguing.

In this work, we found that temperature shifts drastically changed the subcellular localization of exosome subunits. Upon cold-warm shock, exosome subunits were depleted from the nucleoli and accumulated in the nucleoplasm. The nucleolar localization of exosome subunits was gradually restored when animals were shifted back to normal growing temperature. We performed forward genetic screening to search for factors that are required for cold-warm shock-induced exosome translocation and isolated ZTF-7, a zinc finger protein that is an ortholog of human ZNF277. Furthermore, we revealed that ZTF-7 interacted with RPS-2 by immunoprecipitation assay followed by mass spectrometry. RPS-2 is one of the small ribosomal subunit proteins and participates in pre-rRNA processing. A partial depletion of RPS-2 by RNAi blocked the cold-warm stimuli-induced depletion of exosome subunits from the nucleoli. We concluded that the ZTF-7/RPS-2 complex mediates a new cold-warm response pathway in *C. elegans*.

## Results

### Cold-warm shock induced the depletion of the RNA exosome complex from the nucleoli

In *C. elegans*, exosome subunits, including EXOS-1, EXOS-2 and EXOS-10, are ubiquitously expressed in all cells and enriched in nucleoli [25]. Knocking down a number of rRNA processing factors in *C. elegans* by RNAi induced dramatic depletion of EXOS-10 from the nucleoli but enrichment in the nucleoplasm and accumulation of a class of antisense ribosomal siRNAs (risiRNAs) [25]. Meanwhile, cold-warm shock substantially increased risiRNA levels [30].

To test whether cold-warm shock could deplete exosome subunits from the nucleoli, animals expressing GFP::EXOS-10 were cultured at 20˚C until the L3/L4 stage, transferred to 4˚C and grown for 3 hours, and transferred back to 20˚C to recover (Fig 1A). The subcellular localization of GFP::EXOS-10 was visualized under fluorescence microscope. RRP-8 is a nucleolar protein and is used as a nucleolar marker [31].

Growth at 4˚C for 3 hours per se did not alter the nucleolar localization of EXOS-10. However, after recovery at 20˚C for 0.5 hour, EXOS-10 exhibited dramatic depletion from the nucleoli but enrichment in the nucleoplasm (Fig 1B). When recovered at 20˚C for 5 hours, EXOS-10 regained nucleolar localization in the nucleoli. The nucleolar marker RRP-8::mCherry was retained in the nucleoli upon cold-warm treatment (Fig 1B). We performed a time course assay of 4˚C cold treatment followed by 20˚C recovery. Cold shock for at least 2 hours and recovery for 0.5 hour absolutely depleted GFP::EXOS-10 from the nucleolus (Fig 1C). Similarly, we also

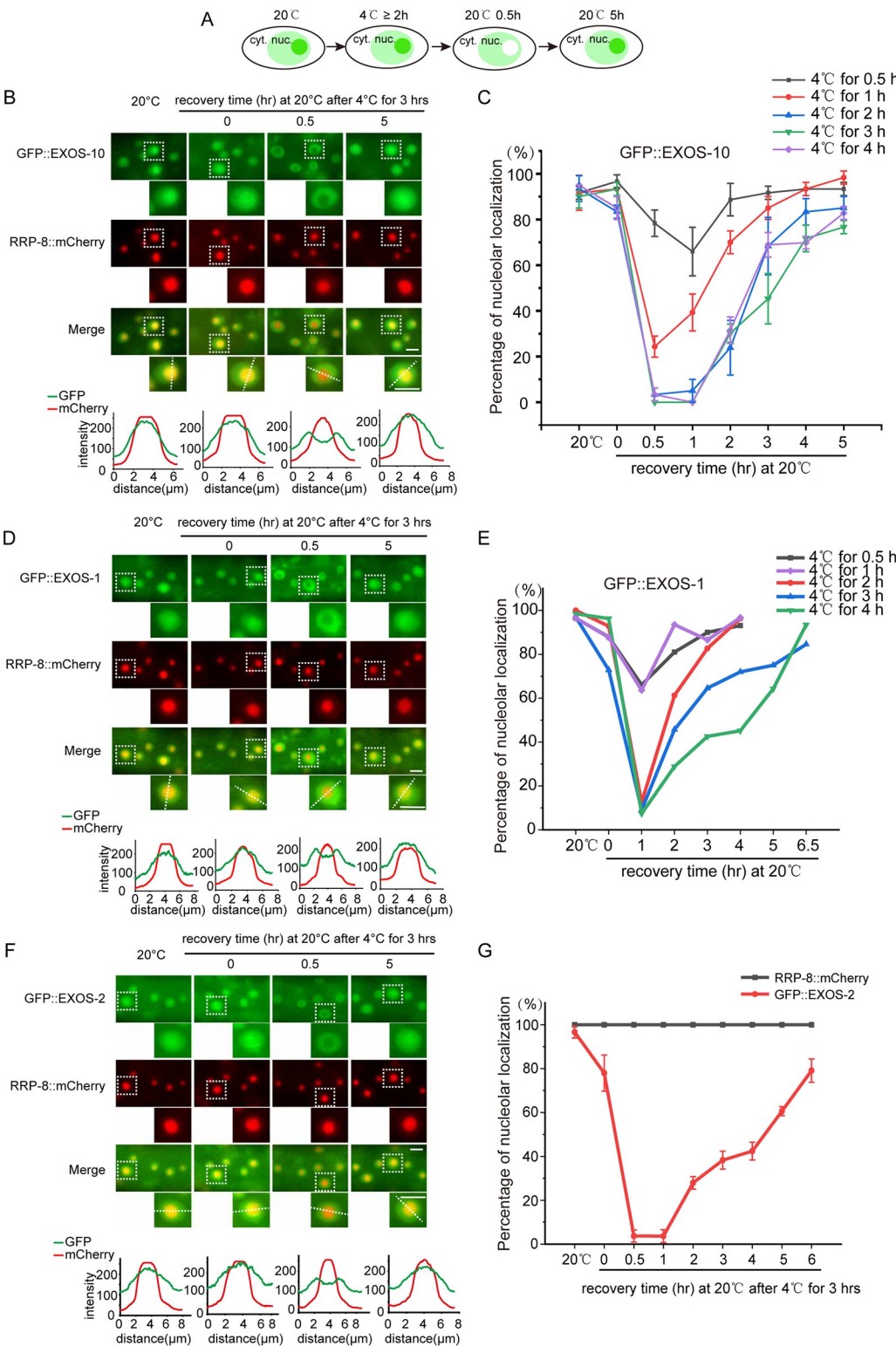

**Fig 1. Cold-warm induced depletion of the RNA exosome complex from the nucleoli.** (A) The model shows the subcellular localization of EXOS-10 after cold-warm shock. cyt., cytoplasm; nuc., nucleus. (B) Images show epidermal cells of the indicated L3/L4 stage animals expressing GFP::EXOS-10 and RRP-8::mCherry after 4˚C for 3 hours. Animals were shifted to 4˚C for 3 hours followed by recovery at 20˚C for 0.5 hour and 5 hours. The fluorescence intensity indicated by dashed lines was measured by ImageJ. Scale bars, 5 μm. (C) The curve graph represents the percentage of

animals with nucleolus-enriched EXOS-10 in epidermal cells at different 4°C treatment durations. n ≥ 20 animals. Nucleolar localization is scored as positive if the observed protein showed nucleolar enrichment in more than half of the epidermal cells in one animal. In each group, more than 20 animals were counted. (D) Images show epidermal cells of the indicated L3/L4 stage animals expressing GFP::EXOS-1 and RRP-8::mCherry after 4°C for 3 hours. The fluorescence intensity indicated by dashed lines was measured by ImageJ. Scale bars, 5 μm. (E) The curve graph represents the percentage of animals with nucleolus-enriched EXOS-1 in epidermal cells at different 4°C treatment durations. n ≥ 20 animals. (F) Images show epidermal cells of the indicated L3/L4 stage animals expressing GFP::EXOS-2 and RRP-8:: mCherry after 4°C for 3 hours. The fluorescence intensity indicated by dashed lines was measured by ImageJ. Scale bars, 5 μm. (G) The curve graph represents the percentage of animals with nucleolus-enriched EXOS-2 and RRP-8 in epidermal cells at 4°C for 3 hours. n ≥ 20 animals.

observed the depletion of EXOS-1 and EXOS-2 from the nucleoli after 4°C culture for 3 hours followed by recovery at 20°C for 0.5 hour (Fig 1D–1G). EXOS-1 and EXOS-2 regained nucleolar localization after recovery at 20°C for 5 hours (Fig 1D–1G). In contrast to other exosome subunits, DIS-3 was the only subunit constantly enriched in the nucleoplasm at 20°C, and its localization was not altered upon cold-warm shock (Fig 2A). From now on, a standard cold-warm shock was referred to 4°C culture for 3 hours followed by recovery at 20°C for 0.5 hour until otherwise indicated.

To determine whether cold-warm shock disrupts nucleolar structure, we investigated the subcellular localization of a number of nucleolar proteins. In addition to RRP-8, three other nucleolar proteins, NUCL-1, RBD-1 and C27F2.4, were retained in the nucleoli upon cold-warm shock treatment (Fig 2B). These data suggested that the gross structure of nucleoli was not disrupted after transient cold-warm exposure.

RPOA-2 is a core subunit of RNA polymerase I and is enriched at rDNA loci [25]. Surprisingly, GFP::RPOA-2 was partially mislocalized from the nucleoli after cold-warm shock (Fig 2C).

GLR-3 encodes a kainate-type glutamate receptor and has been reported to sense cold in the peripheral sensory neuron ASER and trigger cold avoidance behavior [11]. We generated a *glr-3(ust166)* allele by CRISPR/Cas9 technology (Fig 2D). In *glr-3(ust166)*;GFP::EXOS-10 animals, cold-warm shock still triggered the depletion of EXOS-10 from the nucleoli (Fig 2E and 2F). ZIP-10 and ASP-17 mediate the cold-warm response and promote survival ability under resource-limiting and thermal stress conditions [16]. We generated *asp-17(ust190)* and *zip-10 (ust192)* alleles, both of which were out-of-frame and likely null alleles (Fig 2D). In *asp-17 (ust190)*;GFP::EXOS-10 and *zip-10(ust192)*;GFP::EXOS-10 animals, cold-warm shock still triggered the depletion of EXOS-10 from the nucleoli (Fig 2E and 2F). These data suggested that there exist additional pathways, other than GLR-3 or ASP-17/ZIP-10 axis, to response to cold or cold-warm stress.

## Forward genetic screening identified ZTF-7, which mediates cold-warm shock-induced exosome depletion from nucleoli

To further understand the mechanism of the cold-warm response, we conducted a forward genetic screening to search for factors required for cold-warm shock-induced GFP::EXOS-10 translocation. We chemically mutagenized GFP::EXOS-10 animals and searched for mutants in which EXOS-10 failed to translocate from the nucleoli to the nucleoplasm upon cold-warm exposure by clonal screening (Fig 3A). From two thousand haploid genomes, we isolated one mutant allele, *ust117* (Fig 3B).

We mapped *ust117* to the open reading frame F46B6.7 by genome resequencing. F46B6.7 encodes the protein ZTF-7, which is predicted to be an ortholog of human ZNF277 (Fig 3C and 3D). ZTF-7 is an evolutionarily conserved protein with clusters of C2H2-type zinc finger domains whose function remains unknown. ZNF277 deletion may contribute to the risk of language impairments [32]. ZTF-7 contains five zinc finger domains. In the *ust117* allele, the

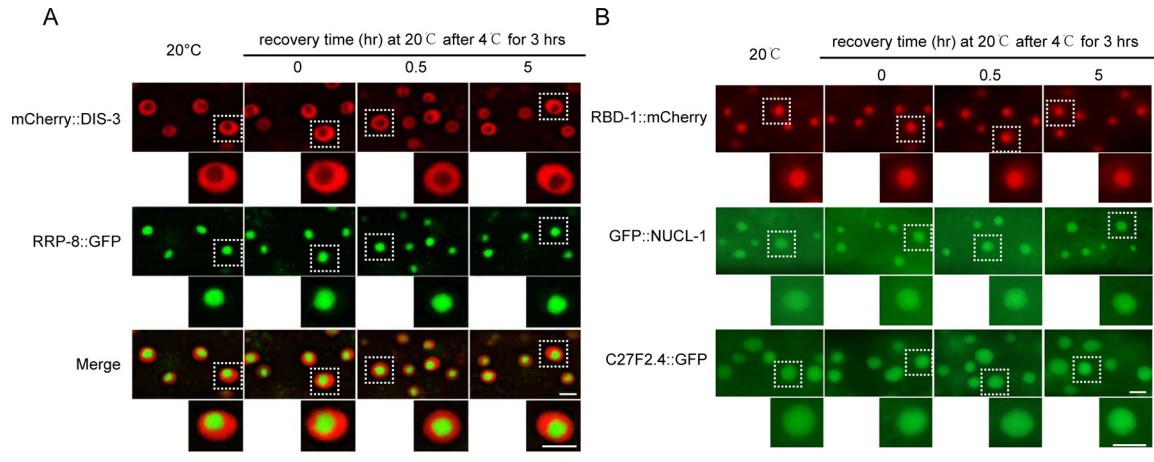

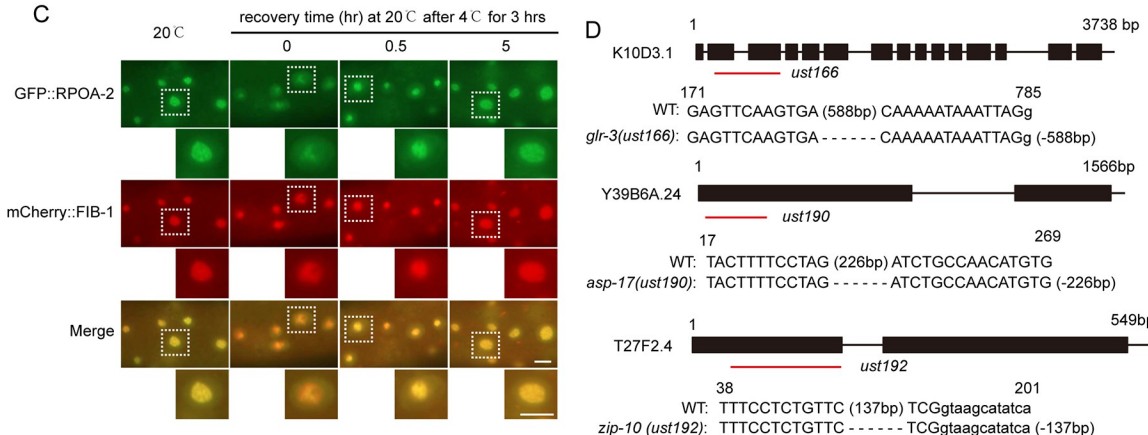

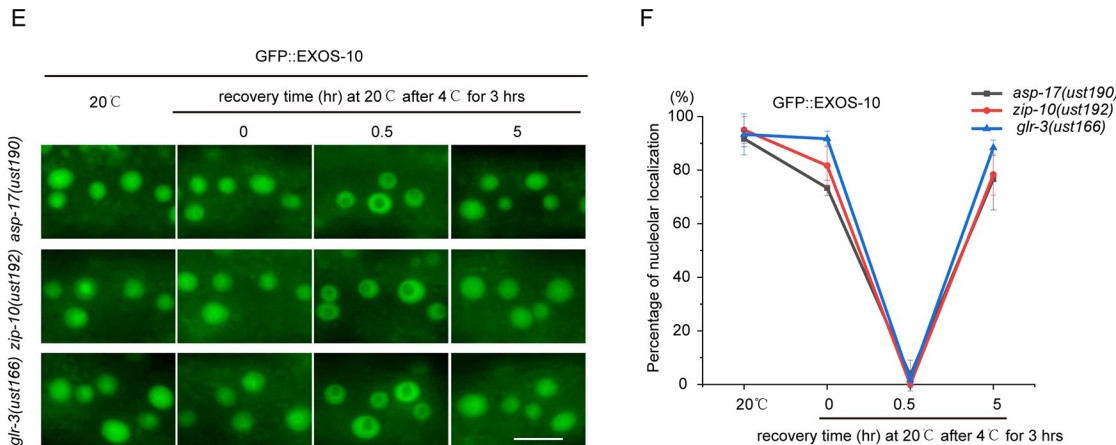

**Fig 2. Cold-warm shock-induced subcellular translocation of the exosome complex implies a unique temperature sensing pathway.** (A) Images show epidermal cells of the indicated L3/L4 stage animals expressing mCherry::DIS-3 and RRP-8::GFP after 4°C for 3 hours. Scale bars, 5 μm. (B) Images of epidermal cells expressing RBD-1::mCherry, GFP::NUCL-1 or C27F2.4::GFP after 4°C for 3 hours at the L3/L4 stage. Scale bars, 5 μm. (C) Images of epidermal cells expressing GFP::RPOA-2 and mCherry::FIB-1 after 4°C for 3 hours at the L3/L4 stage. Scale bars, 5 μm. (D) Alleles of *glr-3(ust166)*, *asp-17(ust190)* and *zip-10(ust192)* used in this study. The deletion alleles of

*ust166*, *ust190* and *ust192* were constructed by CRISPR/Cas9 technology. All three alleles were likely null alleles. (E) Images of the localization of EXOS-10 in the indicated animals after 4˚C for 3 hours. Images were taken from epidermal cells. Scale bars, 10 μm. (F) The curve graph represents the percentage of animals with nucleolus-enriched EXOS-10 in epidermal cells at 4˚C for 3 hours. n ≥ 20 animals.

conserved amino acid Glu437 was mutated to a stop codon. We generated two additional deletion alleles, *ztf-7(ust118)* and *ztf-7(ust119)*, by CRISPR/Cas9 technology (Fig 3B, 3C and 3E). Both alleles are likely null alleles and block cold warm shock-induced EXOS-10 translocation (Fig 3B). The *ztf-7* mutants have a brood size similar to that of wild-type N2 animals at 20˚C but have a much lower number of progeny at 25˚C (Fig 3F and 3G). We assayed the lifespan of *ztf-7* mutants. The *ztf-7* mutation did not pronouncedly affect the lifespan of animals at both 20˚C and 25˚C (Fig 3H and 3I).

To test whether ZTF-7 is involved in cold tolerance response, we grew L3-stage N2 and *ztf-7* animals at 3˚C for 48 hours and transferred back to 20˚C for 8 hours. Both *ztf-7(ust118)* and *ztf-7 (ust119)* mutants exhibited higher survival rate than N2 animals after the cold-warm stress (Fig 3J), suggesting that ZTF-7 limits the ability of animals to tolerate cold environment or cold-warm stress.

We conducted a time course recovery assay after 3 hours of 4˚C exposure. In *ztf-7* mutants, EXOS-10 was constantly retained in the nucleoli upon cold-warm exposure (Fig 4A and 4B). Similarly, in *ztf-7* mutants, cold-warm shock failed to translocate EXOS-1 and EXOS-2 from the nucleoli to the nucleoplasm (Fig 4C and 4D). We ectopically generated a single-copy mCherry-fused ZTF-7 transgene driven by the *ztf-7* promoter via CRISPR/Cas9 technology on chromosome I and introduced it back into *ztf-7(ust117);gfp::exos-10*. The mCherry::ZTF-7 transgene restored the cold-warm shock-induced EXOS-10 translocation (Fig 4E).

Our previous work showed that low temperature increased the expression of risiRNAs and induced the translocation of NRDE-3 from the cytoplasm to the nucleus and nucleoli [30]. We introduced the *ztf-7* mutation into *eri-1(mg366);GFP::NRDE-3* animals. However, the *ztf-7* mutation did not alter the cold-warm shock-induced cytoplasm-to-nucleus translocation of NRDE-3 (Fig 4F).

EXOS-10 in *C. elegans* includes 876 amino acids and three distinct functional domains (Fig 5A). The N-terminal PMC2NT domain binds to the protein cofactor Rrp47, which is necessary for Rrp6 stability and assists the recruitment of the TRAMP complex in yeast [33–35]. The EXO domain contains the 3' to 5' exoribonuclease active site. Both the EXO and HRDC domains constitute the Rrp6 catalytic module (CAT) [36]. To explore whether the correct subcellular localization of EXOS-10 depends on domain integrity, we mutated residues within the EXO domain (D303N, E305Q) known to be necessary for exoribonuclease activity [termed EXOS-10(*EXO)] and generated a GFP-tagged EXOS-10(*EXO) transgene on chromosome IV. Meanwhile, we constructed GFP::EXOS-10(ΔEXO) and GFP::EXOS-10(ΔHRDC) transgenic animals by deleting the EXO domain and HRDC domain, respectively. All of the mutated EXOS-10 transgenes were localized in the nucleoplasm when grown at 20˚C (Fig 5A). In the *ztf-7* mutant, these EXOS-10 mutants were retained in the nucleoplasm at 20˚C as well, suggesting that a second mechanism is involved in EXOS-10 translocation.

Previously, we found that the depletion of *fib-1*, *M28.5*, *mtr-4*, *nol-56*, *T22H9.1* and *rrp-8* by RNAi mislocalized EXOS-10 from the nucleoli to the nucleoplasm [25]. The *ztf-7* mutation could not block EXOS-10 mislocalization either. Both GFP::EXOS-10 and *ztf-7(ust119);*GFP::EXOS-10 animals exhibited similar nucleolar depletion of EXOS-10 upon knocking down *fib-1*, *M28.5*, *mtr-4*, *nol-56*, *T22H9.1* and *rrp-8* (Fig 5B). We partially knocked down *fib-1*, *M28.5*, *mtr-4*, *nol-56*, *T22H9.1* and *rrp-8* for 12 hours by RNAi, but EXOS-10 still translocated to the nucleoplasm (Fig 5C–5E).

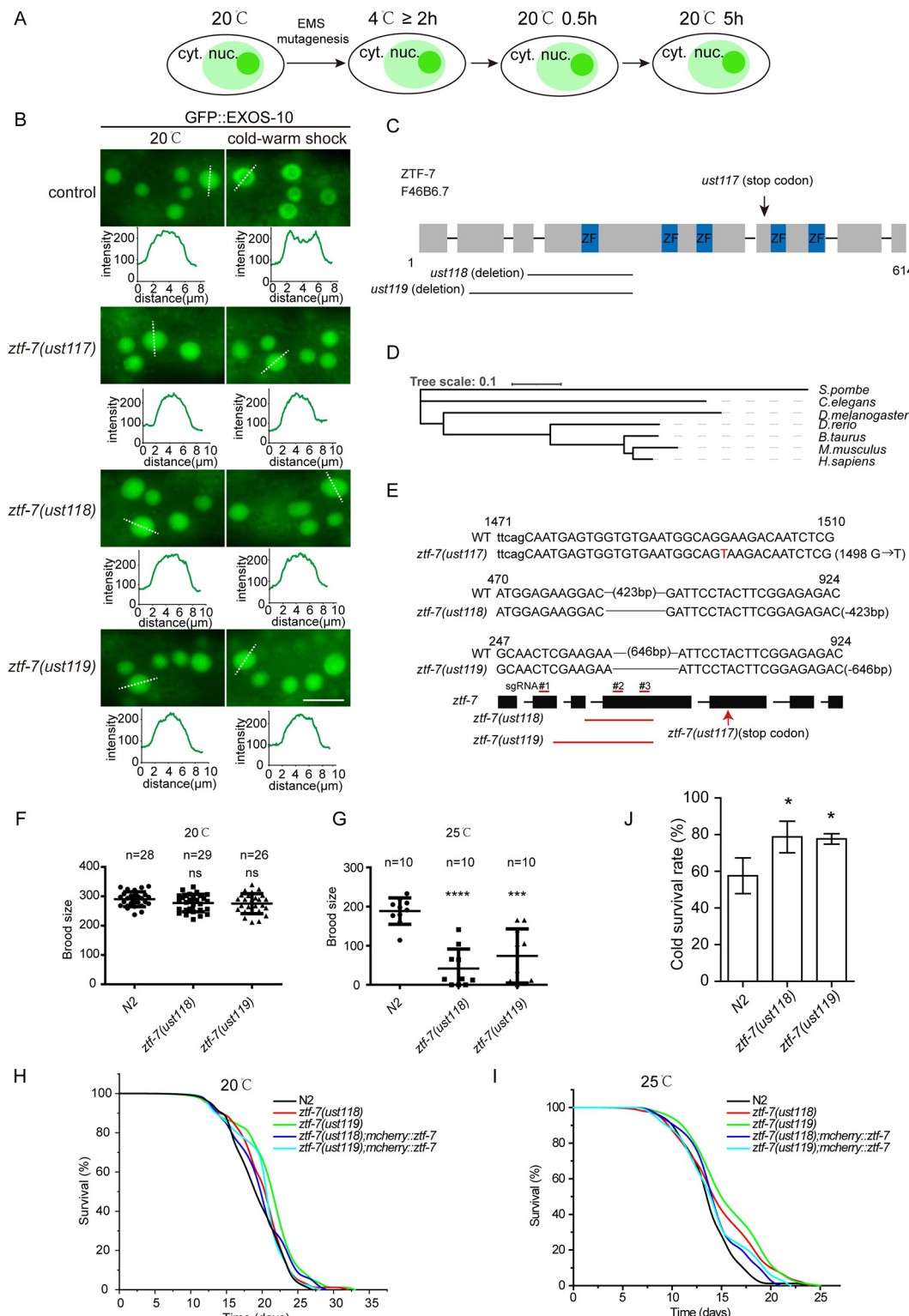

**Fig 3. ZTF-7 is required for the cold-warm shock-induced nucleolar depletion of the RNA exosome complex.** (A) Schematic showing forward genetic screening to search for factors required for cold-warm shock-induced EXOS-10 translocation. cyt., cytoplasm; nuc., nucleus. (B) Images of the localization of EXOS-10 at 20°C and after cold-warm shock at 4°C for 3 hours followed by recovery at 20°C for 0.5 hour. Images were taken from epidermal cells. The fluorescence intensity indicated by dashed lines was measured by ImageJ. Scale bars, 10 μm. (C) Schematic of the ZTF-7 domain structure.

(D) The evolutionary tree of ZTF-7. (E) Alleles of *ztf-7* used in this study. The deletion alleles *ust118* and *ust119* were constructed by CRISPR/Cas9 technology. All three alleles were likely null alleles. (F) Brood size of *ztf-7* mutants grown at 20˚C. Two-tailed Student's t test. Data are presented as the mean ± s.d.; n ≥ 26 animals; ns, not significant. (G) Brood size of *ztf-7* mutants grown at 25˚C. Two-tailed Student's t test. Data are presented as the mean ± s.d.; n = 10 animals; *** P < 0.001, **** P < 0.0001. (H, I) Life span of *ztf-7* mutants grown at 20˚C and 25˚C, respectively. n ≥ 50. (J) Cold survival experiments of N2 and *ztf-7* mutants. Synchronized L3 animals were grew at 3˚C for 48 hours. Animals were then transferred back to 20˚C and grew for 8 hours and scored. Data are presented as the mean ± s.d.; n = 3, Two-tailed Student's t test, * P < 0.05.

## ZTF-7 interacts with RPS-2

To understand how ZTF-7 regulates cold-warm shock-induced exosome translocation, we generated a GFP-3xFLAG-tagged ZTF-7 transgene at the endogenous locus by CRISPR/Cas9 technology. LMN-1 is an ortholog of human lamin A/C and is localized at the nuclear envelope. To our surprise, ZTF-7 was constantly localized in the cytoplasm, and cold-warm shock did not alter the subcellular localization of ZTF-7 (Fig 6A).

We immunoprecipitated ZTF-7, resolved it by SDS–PAGE and stained it with coomassie blue dye. We cut off the strongest band at approximately 28 kD from the gel, subjected it to mass spectrometry, and identified RPS-2 (Fig 6B). RPS-2 is one of the proteins of the small ribosome subunits and participates in pre-rRNA processing and nuclear export of pre-40S subunits in fission yeast [37]. Strikingly, human ZNF277 binds extraribosomal uS5(RPS2) as well, suggesting a conserved function of ZTF-7/RPS-2 complex [38].

To confirm the ZTF-7/RPS-2 interaction in *C. elegans*, we knocked in a 3xHA tag onto the *rps-2* gene. We conducted a co-IP experiment in 3xHA::RPS-2;ZTF-7::GFP::3xFLAG animals and revealed the interaction between ZTF-7 and RPS-2 in vivo (Fig 6C). However, we failed to detect significant effects of cold-warm shock on the ZTF-7/RPS-2 interaction. Knocking down *rps-2* by RNAi induced a dramatic decrease in ZTF-7 expression, as shown by fluorescent microscope (Fig 6D and 6E). Notably, the depletion of *rps-2* by RNAi resulted in a significant reduction of the mRNA levels of *ztf-7* to roughly 70% of that of control animals (Fig 6F), suggesting that RPS-2 may promote the transcription of ZTF-7. Western blotting further confirmed the reduction of ZTF-7::GFP::3xFLAG protein after *rps-2* RNAi (Fig 6G). Knocking down other subunits of the ribosome by RNAi did not pronouncedly alter the expression level of ZTF-7, suggesting that RPS-2 is specifically required to maintain ZTF-7 levels (Fig 6H and 6I).

## Small ribosomal subunit proteins are required for cold-warm shock-induced exosome translocation

To test whether RPS-2 is required for the cold-warm shock response, we knocked down a number of *rps* genes by RNAi. Strikingly, long-term exposure to *rps* dsRNAs at 20˚C depleted GFP::EXOS-10 from the nucleoli but enriched it in the nucleoplasm. The *ztf-7* mutation could not block EXOS-10 mislocalization either, suggesting a process not requiring the presence of ZTF-7 (Fig 7A).

Then, we tried to feed animals *rps-0*, *rps-2* and *rps-12* dsRNAs for a shorter time at 20˚C and subsequently subjected the animals to a cold-warm shock assay. Knocking down *rps-0*, *rps-2* and *rps-12* for 4, 6, 8 or 12 hours did not markedly change the nucleolar localization of EXOS-10 at 20˚C. However, 6, 8 or 12 hours of RNAi of *rps-0*, *rps-2* and *rps-12* significantly blocked cold-warm shock-induced EXOS-10 translocation (Fig 7B–7F).

Taken together, these data suggest that small ribosomal subunit proteins are involved in cold-warm shock-induced exosome translocation.

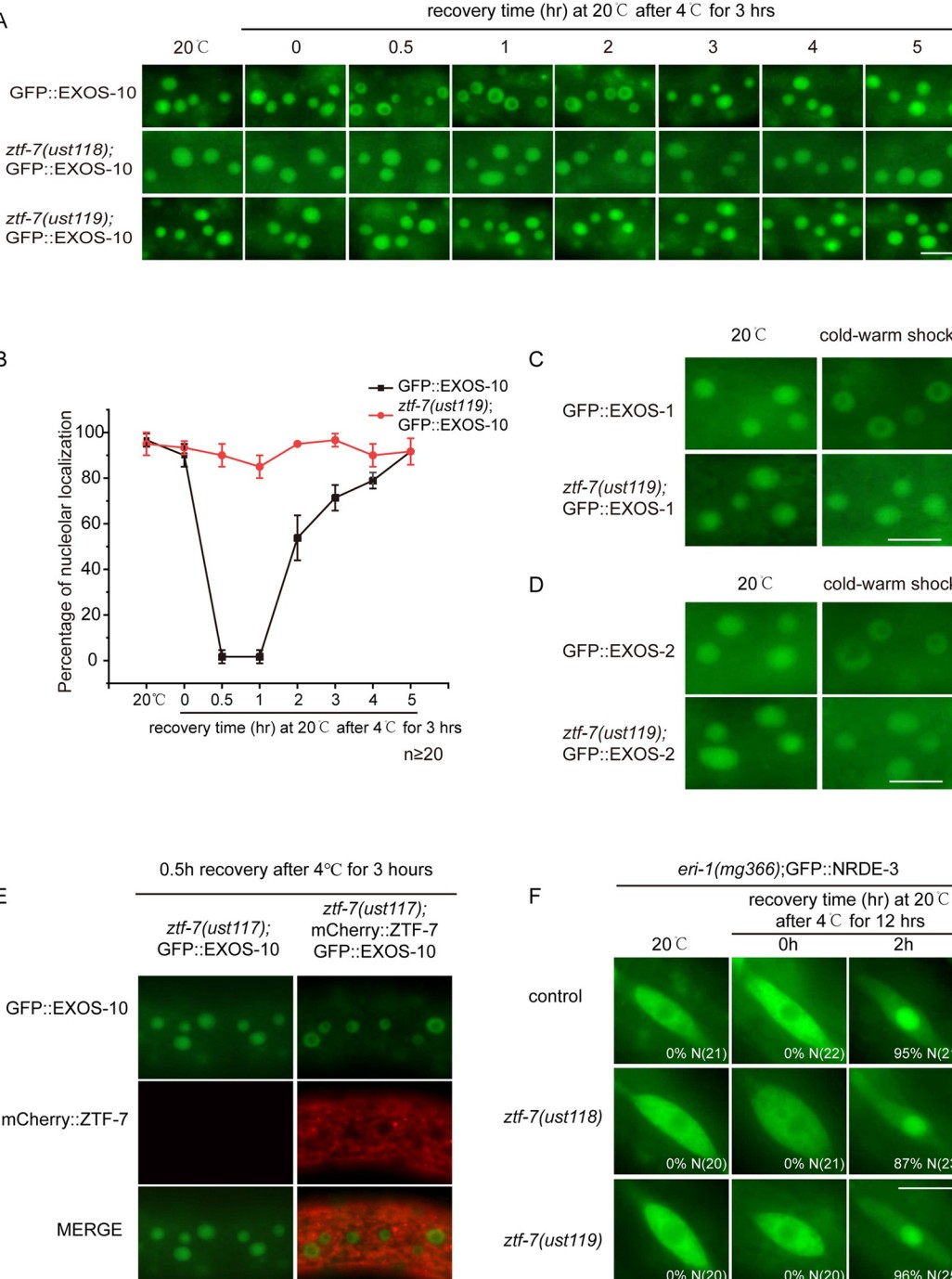

**Fig 4. ZTF-7 is specifically required for cold-warm shock-induced exosome translocation.** (A) Images of the localization of EXOS-10 at 20°C and after cold-warm shock at 4°C for 3 hours followed by recovery at 20°C for different times. Images were taken from epidermal cells. Scale bars, 10 μm. (B) The curve graph represents the percentage of animals with nucleolus-enriched EXOS-10 in epidermal cells after 4°C for 3 hours. n ≥ 20 animals. (C, D) Images of the localization of EXOS-1 (C) and EXOS-2 (D) at 20°C and after 4°C for 3 hours followed by recovery at 20°C for 0.5 hour. Scale bars, 10 μm. (E) Images of epidermal cells of *ztf-7(ust117);gfp::exos-10;mCherry::ztf-7* animals at L3/L4 stage. *mCherry::ztf-7* rescued the cold-warm shock-induced translocation of GFP::EXOS-10 in *ztf-7(ust117)* animals. Scale bars, 10 μm. (F) Images show seam cells of the indicated L4 stage animals expressing GFP::NRDE-3. Numbers indicate the percentage of animals with nucleus-enriched NRDE-3 in seam cells (% N). The number of scored animals is indicated in parentheses. Scale bars, 10 μm. Worms were treated at 4°C for 12 hours and then grown at 20°C for 2 h, followed by counting and imaging.

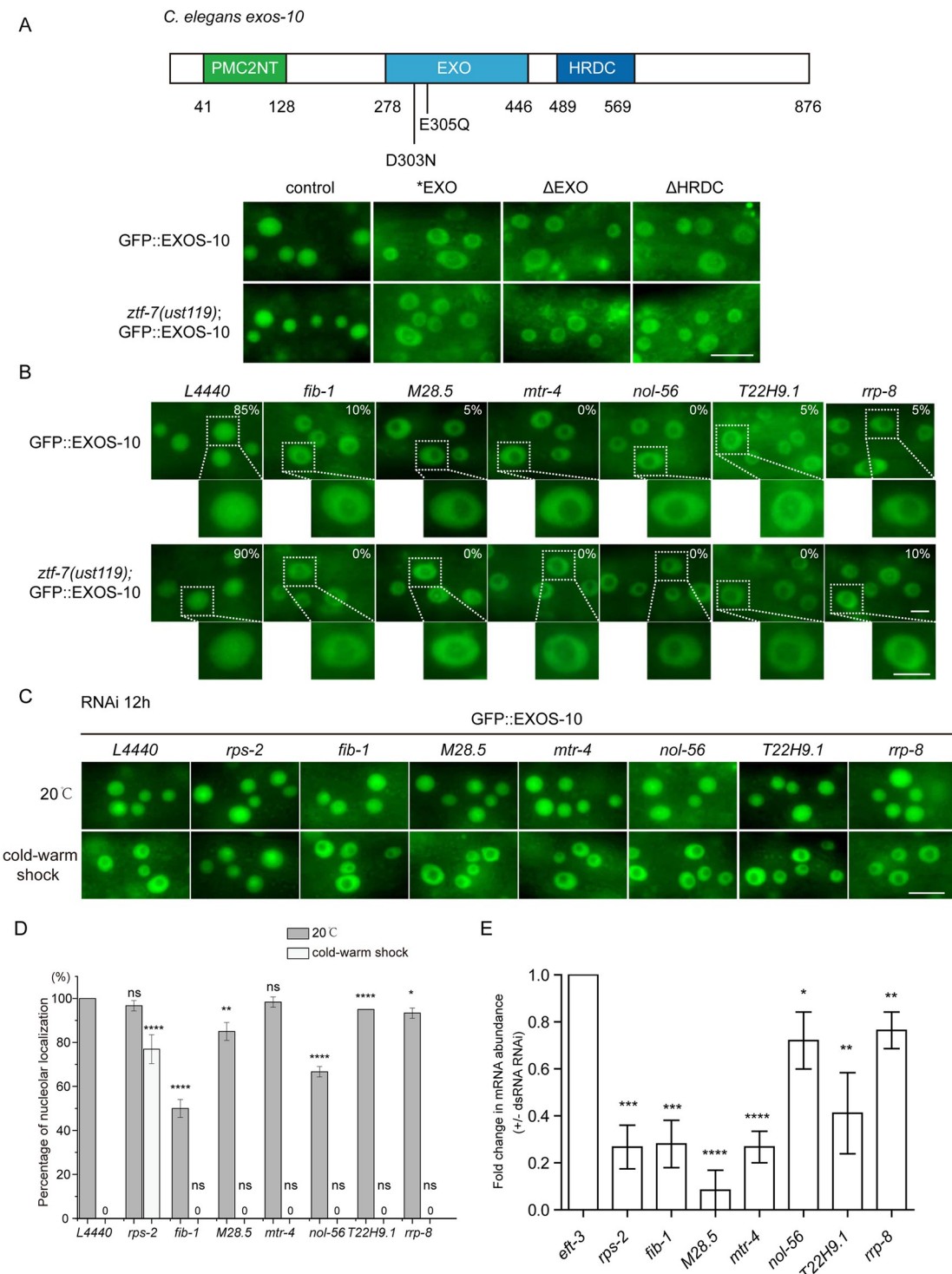

**Fig 5. Disruption of EXOS-10 exoribonuclease domains mislocalized EXOS-10 from the nucleolus to the nucleoplasm.** (A) (Top) Domain structure of EXOS-10. (Bottom) Images of the animals expressing the indicated EXOS-10 transgenes grown at 20˚C without cold-warm shock. Images were taken from epidermal cells. Scale bars, 10 μm. EXOS-10 (*EXO): D303N;E305Q. GFP::EXOS-10(ΔEXO) and GFP::EXOS-10(ΔHRDC) transgenic animals by deleting the EXO domain and HRDC domain, respectively. (B) Images of epidermal cells of animals expressing GFP::EXOS-10 after treatment with RNAi targeting the indicated genes. Bleached embryos carrying EXOS-10 transgenes were grown on RNAi plates for 48 h before being photographed. The percentage of animals with nucleolar-localized GFP::EXOS-10 is indicated. n ≥ 20. Scale bars, 5 μm. Animals were cultured at 20˚C. (C) Images of epidermal cells of animals expressing GFP::EXOS-10 after treatment with RNAi targeting the indicated genes

for 12 hours. Animals were cold-warm shocked at 4˚C for 3 hours followed by recovery at 20˚C for 0.5 hour, and then the images of EXOS-10 were taken at 20˚C. Scale bars, 10 μm. (D) Quantification of the nucleolar localization of GFP::EXOS-10. Nucleolar localization is scored as positive if the observed protein showed nucleolar enrichment in more than half of epidermal cells in one animal. n ≥ 20 animals. Two-tailed Student's t test. ns, not significant. *P < 0.05, **P < 0.01, **** P < 0.0001. (E) qRT-PCR assay of the indicated mRNAs after RNAi for 12 hours. mean ± s.d. n = 3. Two-tailed Student's t test. * P < 0.05; ** P < 0.01, *** P < 0.001, **** P < 0.0001.

## Discussion

Here, we showed cold-warm exposure-induced exosome translocation from the nucleolus to the nucleoplasm. We identified a conserved ZTF-7/RPS-2 complex that is required for this cold-warm response in *C. elegans*. Partial knockdown of RPS proteins inhibited cold-warm exposure-induced exosome translocation. Our results may reveal a new mechanism by which animals respond to temperature alterations in the environment.

The RNA exosome is a conserved ribonuclease complex that is involved in RNA degradation in both the nucleus and cytoplasm. In *S. cerevisiae*, exosome complexes containing the Rrp6 subunit are exclusively localized in the nucleus, whereas exosomes with Dis3 can be found throughout the cell. In human cells, Dis3 is excluded from the nucleolus [20]. Previously, we surveyed the localization of RNA exosome complex subunits in *C. elegans* and found that one of the core catalytic active subunits, EXOS-10, is enriched in the nucleolus, while the other subunit, DIS-3(Rrp44), is mainly localized in the nucleoplasm [25]. The exosome complex associates with compartment-specific adaptors to engage in diverse processes. In human cells, in the nucleoplasm, the nuclear exosome targeting (NEXT) complex and the poly(A) tail eXosome targeting (PAXT) complex efficiently promote the degradation of short capped transcripts, such as promotor upstream transcripts (PROMPTs), enhancer RNAs (eRNAs) and prematurely terminated (pt) transcripts [39–42]. Exosome translocated from the nucleolus to nucleoplasm upon cold-warm exposure, and restored the nucleolar accumulation during further recovery. The function of this exosome translocation is unclear. We speculated that cold-warm exposure may result in the accumulation of erroneous RNAs in certain subnuclear compartments and induced the translocation. Further investigations are required to test whether forced mislocalization of exosome factors could trigger distinct biological responses.

ZTF-7 is an evolutionarily conserved protein with clusters of C2H2-type zinc finger domains. In humans, ZNF277 forms a stable extraribosomal complex with the 40S ribosomal protein uS5 (RPS2) in a cotranslational manner [38]. In *C. elegans*, ZTF-7 interacts with RPS-2 as well. Here, we showed that the interaction between RPS-2 and ZTF-7 are two folds. First, ZTF-7 associates with RPS-2 via protein-protein interaction. Second, RPS-2 likely promotes the transcription of *ztf-7*, yet the mechanism is still unclear. The ZTF-7/RPS-2 complex is required for the cold-warm-induced nucleolar depletion of RNA exosomes. In human cells, ZNF277 is localized in both the nucleus and cytoplasm [38], while ZTF-7 in *C. elegans* is mainly localized in the cytoplasm. The subcellular localization of ZTF-7 is unaffected by cold-warm shock. How the conserved ZTF-7/RPS-2 complex senses temperature shifting and induces the response of nucleolar exosome proteins is intriguing. Whether the ZTF-7/RPS-2 complex aids temperature sensing in human cells requires investigation.

A recent proteomics approach identified that human ZNF277 binds the RNA sequences of p38–MAPK14 mRNA [43], thus raising the possibility that ZTF-7 can bind RNAs in the cytoplasm. RPS-2 localized in both the cytoplasm and nucleus, including the nucleoplasm and nucleolus. The synthesis and processing of rRNA, as well as the assembly between rRNAs and ribosomal proteins, occur in the nucleolus [37]. Rps2 is required for efficient processing of the 32S pre-rRNA at site A2, and most pre-40S subunits are rapidly degraded in the absence of Rps2 in the

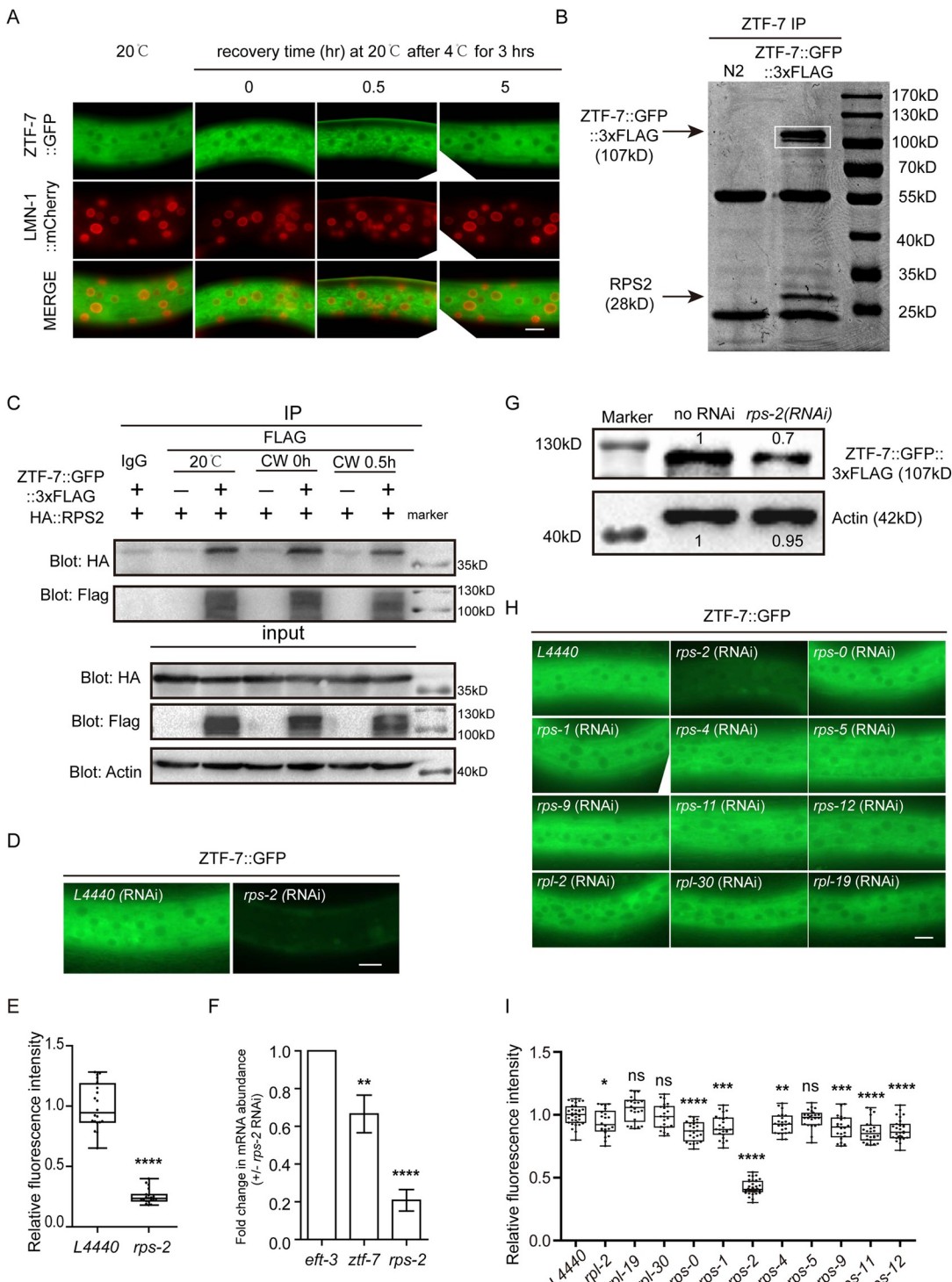

**Fig 6. ZTF-7 interacts with RPS-2.** (A) Images of somatic cells of L4 animals expressing ZTF-7::GFP after 4°C for 3 hours. Scale bars, 10 μm. (B) ZTF-7::GFP::3xFLAG was immunoprecipitated with an anti-FLAG antibody, resolved by SDS–PAGE and stained with Coomassie blue dye. ZTF-7::GFP::3xFLAG and RPS-2 bands are indicated by arrows. In the white box, both bands are ZTF-7::GFP::3xFLAG, as identified by mass spectrometry separately. (C) Co-IP assay in L3/L4 stage animals expressing ZTF-7::GFP::3xFLAG and HA::RPS2 using anti-FLAG antibody. RPS2 was detected by an anti-HA antibody. (D) Images of somatic cells of animals expressing ZTF-7::GFP after treatment with RNAi targeting *rps-2*. Scale bars, 10 μm. Animals were cultured at 20°C. (E) Box plots showed relative fluorescence intensities of ZTF-7::GFP::3xFLAG after knocking down *rps-2* by RNAi. n = 20. Two-tailed Student's t test. **** P < 0.0001. (F) Quantification of *ztf-7* mRNA levels after RNAi *rps-2*. mean ± s.d. n = 3. Two-

tailed Student's t test. ** $P < 0.01$; **** $P < 0.0001$. (G) The ZTF-7 protein level was measured by western blotting using L3/L4 stage animals. Band density of ZTF-7::GFP::3xFLAG was quantified by ImageJ. (H) Images of somatic cells of animals expressing ZTF-7::GFP after treatment with RNAi targeting the indicated genes. Bleached embryos were grown on RNAi plates for 48 h before being photographed. Scale bars, 10 μm. Animals were cultured at 20°C. (I) Box plots showed relative fluorescence intensities of ZTF-7::GFP::3xFLAG after knocking down the indicated genes by RNAi. $n \geq 22$. Two-tailed Student's t test. ns, not significant, * $P < 0.05$; ** $P < 0.01$, *** $P < 0.001$, **** $P < 0.0001$.

nucleolus in fission yeast [37]. ZNF277 is speculated to function as a chaperone and protect and escort free uS5 to 40S precursors [38]. We hypothesize that cold-warm shock may diminish the function of the ZTF-7/RPS-2 complex that leads to a deficiency of rRNA maturation and ribosome assembly, which subsequently induces the translocation of the exosome complex from the nucleolus to the nucleoplasm. However, the mechanism of the latter is unknown.

Temperature change is an inevitable environmental condition for all kinds of living organisms. Cold stimuli, particularly noxious cold, are not only life-threatening but also cause severe tissue damage and evoke pain in animals and humans [44]. How to sense temperature change and respond accordingly is very important for organism survival. For *C. elegans*, wild-type worms grown at 15°C can survive at a temperature of 2°C, whereas those grown at 20°C or 25°C cannot [8]. The growth temperatures under laboratory conditions are typically 15–25°C. Worms at 15–16°C live longer than those at 25°C [15,45,46]. TRPA-1, a cold-sensitive TRP channel, detects temperature drops in the environment to extend lifespan at the adult stage; this effect requires cold-induced, TRPA-1-mediated calcium influx and a calcium-sensitive PKC that signals to the transcription factor DAF-16/FOXO [15,47]. The adiponectin receptor PAQR-2 signaling acts as a regulator linking low temperature with autophagy to extend lifespan in *C. elegans* [48]. Reducing core body temperature increased the life span of mice[49]. The ZIP-10/ASP-17 genetic program mediates the cold-warming response and may have evolved to promote wild-population kin selection under resource-limiting and thermal stress conditions [16]. However, the ZIP-10/ASP-17 genetic program is not required for cold-warm shock-induced exosome translocation.

## Materials and methods

### Strains

Bristol strain N2 was used as the standard wild-type strain. All strains were grown at 20°C unless specified. The strains used in this study are listed in S1 Table.

### Low-temperature treatment

Synchronized embryos were cultivated at 20°C until the L3/L4 stage and then transferred to 4°C and cultured for the indicated hours followed by fluorescence microscope. For the standard cold-warm shock assay, animals were treated at 4°C for 3 hours unless specified.

### Forward genetic screening

To identify factors that can regulate cold-warm shock-induced exosome depletion from nucleoli, we chemically mutagenized GFP::EXOS-10 animals by ethyl methanesulfonate (EMS), followed by clonal screening. The F3 progeny worms at the L3/L4 stage were shifted from 20°C to 4°C for three hours. Mutants were subsequently recovered at 20°C for 0.5 hour and visualized under a fluorescence microscope. Mutants that failed to redistribute EXOS-10 from the nucleolus to the nucleoplasm upon cold-warm shock were selected. We isolated one mutant allele from approximately 2000 haploid genomes. *ztf-7* was identified by genome resequencing.

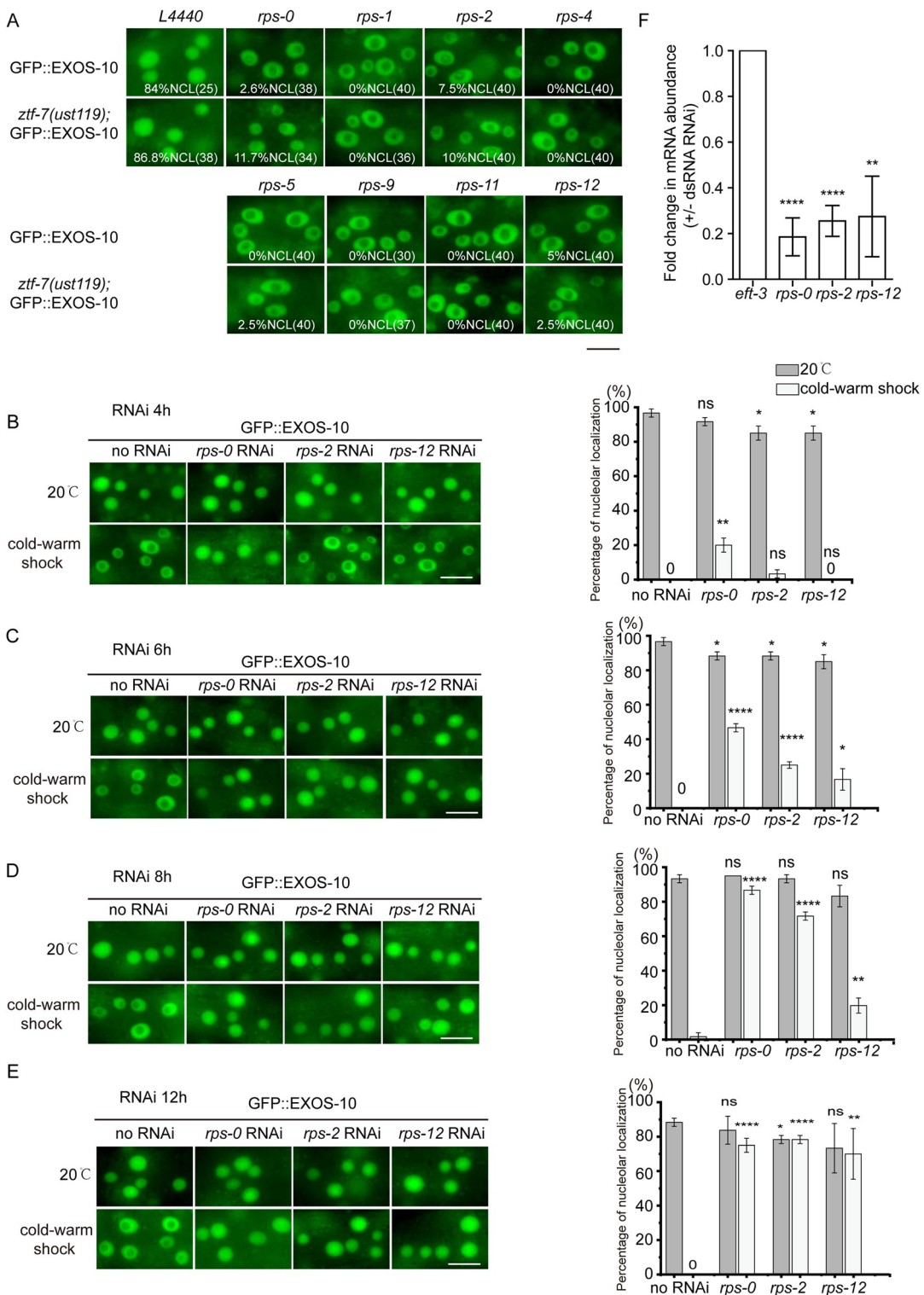

**Fig 7. RPS proteins are required for cold-warm shock-induced RNA exosome translocation.** (A) Images of epidermal cells of animals expressing GFP::EXOS-10 after treatment with RNAi targeting the indicated genes. Bleached embryos carrying EXOS-10 transgenes were grown on RNAi plates for 48 h before being photographed. The percentage of animals with nucleolar-localized GFP::EXOS-10 is indicated (% NCL). The number of scored animals is indicated in parentheses. Scale bars, 10 μm. Animals were cultured at 20°C. (B-E) (Left) Images of epidermal cells of animals expressing GFP::EXOS-10 after treatment with RNAi targeting the indicated genes for 4, 6, 8 or 12 hours. Animals were cold-warm shocked at 4°C for 3 hours

followed by recovery at 20˚C for 0.5 hour, and then the images of EXOS-10 in epidermal cells were taken at 20˚C. Scale bars, 10 µm. (Right) Quantification of nucleolar-localized GFP::EXOS-10. Nucleolar localization is scored as positive if the observed protein showed nucleolar enrichment in more than half of epidermal cells in one animal. n ≥ 20 animals. Two-tailed Student's t test. ns, not significant. * P < 0.05, ** P < 0.01, **** P < 0.0001. (F) qRT-PCR assay of the indicated mRNAs after RNAi for 12 hours. ** P < 0.01, **** P < 0.0001.

## Construction of plasmids and transgenic strains

For in situ transgene ZTF-7::GFP::3xFLAG construction, a GFP::3xFLAG region was PCR amplified with the primers 5'-GGAGGTGGAGGTGGAGCTA-3' and 5′-CTTGTCATCGT CATCCTTGTAA-3′ from the genomic DNA of UAD-2::GFP::3xFLAG. A 1.5 kb homologous left arm was PCR amplified with the primers 5′-GGGTAACGCCAGCACGTGTGGATTCTG ATGATGAGGATTCACGAC-3′ and 5'-ATAGCTCCACCTCCACCTCCCATGAGAGCG TTGAGGTCATC-3'. A 1.5 kb homologous right arm was PCR amplified with the primers 5'-ACAAGGATGACGATGACAAGTAATTTGACATAACTACCTCATCTCAGC-3' and 5'-CAGCGGATAACAATTTCACACATCGTCAAGTGGAGAATCAG-3'. The backbone was PCR amplified from the plasmid pCFJ151 with the primers 5'-CACACGTGCTGGCGTTA CC-3' and 5'-TGTGAAATTGTTATCCGCTGG-3'. All these fragments were joined together by Gibson assembly to form the *ztf-7::gfp::3xflag* plasmid with the ClonExpress MultiS One Step Cloning Kit (Vazyme Biotech, Nanjing, China, Cat. No. C113-01/02). This plasmid was coinjected into N2 animals with three sgRNA expression vectors, *ztf-7*_sgRNA#1, *ztf-7*_sgRNA#2, *ztf-7*_sgRNA#3, 5 ng/µl pCFJ90 and Cas9 II expressing plasmid.

For the *construction of mCherry::ztf-7* in chromosome I, a 2 kb promoter region was amplified with the primers 5′-CCTGTCAATTCCCAAAATACATCACTTACCGTATTCATATT-3′ and 5′-TCTTCACCCTTTGAGACCATTCCCGAAGTTGACATGGTGTA-3′. The *ztf-7* CDS region and 3′ UTR were amplified as a whole fragment with the primers 5′-AGGGAGGTG GAGGTGGAGCTATGTCAACTTCGGGAAGCG-3′ and 5′-TTCAAAGAAATCGCC GACTTTCCATGATCAACATGTCCCA-3′. The *mCherry* coding sequence was amplified from PFCJ90 with 5′-ATGGTCTCAAAGGGTGAAGAAG-3′ and 5′-AGCTCCACCTC CACCTCCCTTATACAATTCATCCATGCCA-3′. The linearized backbone was amplified from pCZGY2727 with primers 5′-AAGTCGGCGATTTCTTTGAA-3′ and 5′-GTATTTTG GGAATTGACAGGG-3′. The transgene was integrated into *C. elegans* chromosome I of strain N2 by CRISPR/Cas9 technology. Primer pairs for constructing sgRNA expression vectors are shown in S2 Table.

## CRISPR/Cas9-mediated gene deletion

Multiple sgRNA-guided chromosome deletion was conducted as previously described [50]. To construct sgRNA expression plasmids, the 20 bp *unc-119* sgRNA guide sequence in the pU6:: unc-119 sgRNA(F + E) vector was replaced with different sgRNA guide sequences. Addgene plasmid #47549 was used to express Cas9 II protein. A plasmid mixture containing 30 ng/µl of each of the three sgRNA expression vectors, 50 ng/µl Cas9 II expressing plasmid, and 5 ng/µl pCFJ90 was coinjected into animals. The deletion mutants were screened by PCR amplification and confirmed by sequencing. The sgRNAs used in this study are listed in S2 Table.

## RNAi

RNAi experiments were conducted as previously described [51]. HT115 bacteria expressing the empty vector L4440 (a gift from A. Fire) were used as controls. Bacterial clones expressing dsRNA were obtained from the Ahringer RNAi library [52] and were sequenced to verify their

identity. All the RNAi experiments were started by placing the synchronized embryos on seeded RNAi plates unless specified.

## IP-MS

Synchronized animals expressing ZTF-7::GFP were cultured at 20°C until the L3/L4 stage. Animals were harvested, resuspended in lysis buffer (25 mM Tris-HCl at pH 7.4, 150 mM NaCl, 10% glycerol, 1% NP-40, Roche Complete EDTA-free protease inhibitor cocktail, 1 mM EDTA) and lysed by sonication. Lysates were precleared and immunoprecipitated with anti-FLAG antibody (Sigma, F1804) for ZTF-7 overnight at 4°C. Antibody-bound complexes were recovered with Protein A/G agarose beads (Thermo Fisher, 53132). The beads were then washed five times with cold lysis buffer, boiled in SDS loading dye for ten minutes at 95°C, and resolved by SDS–PAGE. The protein band was cut off from SDS–PAGE and then subjected to mass spectrometry (Shanghai Applied Protein Technology Co. Ltd).

## Brood size

Worms of the indicated genotypes were cultured on NGM plates at 20°C or 25°C. No less than ten L4 worms were singled to fresh NGM plates, and the number of progenies was scored.

## Lifespan assay

The lifespan assay was performed at 20°C and 25°C as previously described [53]. Briefly, worms were synchronized by placing young adult animals on NGM plates for 4–6 hours and then removed. The hatching day was counted as day one for all lifespan measurements. Worms were transferred every other day to new plates to eliminate confounding progeny. Animals were scored as alive or dead every two days. Worms were scored as dead if they did not respond to repeated prods with a platinum pick. Worms were censored if they crawled off the plate or died from vulval bursting and bagging. For each lifespan assay, 90 worms were used on 3 plates (30 worms/plate).

## Western blotting

L3/L4-stage worms were harvested, washed three times with M9 buffer, boiled in SDS loading dye for ten minutes at 95°C, and then resolved by SDS–PAGE. Proteins were transferred to a Hybond-ECL membrane and blocked with 5% milk-TBST. The membrane was incubated with antibodies overnight at 4°C. The membrane was incubated with secondary antibodies and then visualized.

The antibody dilutions for western blotting were as follows: mouse anti-FLAG antibody (Sigma, F1804), 1:5,000; mouse anti-HA antibody (Proteintech, 66006-2-Ig); rabbit β-actin (Beyotime, AF5003), 1:4,000. The secondary antibodies used were goat anti-mouse (Beyotime, A0216) and goat anti-rabbit (Abcam, ab205718) antibodies.

## Quantitative real-time PCR (qRT-PCR)

RNAs were isolated from L3/L4 animals using a dounce homogenizer (pestle B) in TRIzol solution followed by DNase I digestion (Fermentas) and isopropanol precipitation. RNAs were reverse transcribed via HiScript II Q Select RT SuperMix (Vazyme #R233) with Oligo (dT)23VN primers. cDNAs were quantified with SYBR Green Master Mix (Vazyme, Q111-02) according to the vendor's instructions, using an MyIQ2 machine (Bio-Rad). The primer sequences for cDNA detection are listed in S3 Table. The numbers of replicates are indicated in figure legends.

### Cold tolerance assay

Synchronized L3 animals were transferred to a refrigerated cabinet (BCD-601W; SIEMENS) and grew at 3˚C for 48 hours. Animals were then transferred back to 20˚C and grew for 8 hours and scored. The temperature inside the cabinet was monitored using a digital thermometer.

### Imaging

Images were collected using a Leica DM4 microscope. All images were taken from epidermal cells unless otherwise specified. Nucleolar localization is scored as positive if the observed protein showed nucleolar enrichment in more than half of the epidermal cells in one animal. In each group, more than 20 animals were counted.

### Statistics

The mean and standard deviation of the results are presented in bar graphs with error bars. All experiments were conducted with independent *C. elegans* animals for the indicated number (N) of times. Statistical analysis was performed with the two-tailed Student's t test.

## Supporting information

**S1 Table. Strains used in the work.**
(DOCX)

**S2 Table. Sequences of sgRNAs for CRISPR/Cas9-mediated gene editing.**
(DOCX)

**S3 Table. Sequences of quantitative real-time PCR primers.**
(DOCX)

## Acknowledgments

We are grateful to the members of the Guang lab for their comments. We are grateful to the International *C. elegans* Gene Knockout Consortium and the National Bioresource Project for providing the strains. Some strains were provided by the CGC, which is funded by the NIH Office of Research Infrastructure Programs (P40 OD010440).

## Author Contributions

**Data curation:** Ting Xu, Shimiao Liao, Meng Huang, Xiangyang Chen.

**Formal analysis:** Ting Xu, Shimiao Liao, Chengming Zhu.

**Funding acquisition:** Shouhong Guang.

**Investigation:** Ting Xu, Shimiao Liao, Meng Huang, Xiaona Huang, Qile Jin, Demin Xu.

**Project administration:** Ting Xu, Chuanhai Fu, Xiangyang Chen, Xuezhu Feng, Shouhong Guang.

**Supervision:** Chuanhai Fu, Xiangyang Chen, Xuezhu Feng, Shouhong Guang.

**Validation:** Ting Xu, Shimiao Liao, Chuanhai Fu, Xiangyang Chen, Xuezhu Feng, Shouhong Guang.

**Visualization:** Chuanhai Fu, Xiangyang Chen, Xuezhu Feng, Shouhong Guang.

**Writing – original draft:** Ting Xu.

**Writing – review & editing:** Shouhong Guang.

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
