## [Decision Letter · Decision Letter 0]

17 Nov 2022

Dear Dr guang,

Thank you very much for submitting your Research Article entitled 'A ZTF-7/RPS-2 complex mediates the cold-warm response in C. elegans' to PLOS Genetics.

The manuscript was fully evaluated at the editorial level and by independent peer reviewers. The reviewers appreciated the attention to an important topic but identified some concerns that we ask you address in a revised manuscript.

We therefore ask you to modify the manuscript according to the review recommendations. Your revisions should address the specific points made by each reviewer.

Yours sincerely,

Dengke K Ma

Guest Editor

PLOS Genetics

Gregory P. Copenhaver

Editor-in-Chief

PLOS Genetics

Reviewer's Responses to Questions

**Comments to the Authors:**

Reviewer #1: The authors demonstrate that ZTF-7 is required for RNA exosome depletion from nucleoli upon transient cold-warm exposure in C. elegans. These effects are dependent upon RPS-2, a ribosomal protein of the small subunit. These results establish a novel mechanism by which C. elegans responds to environmental cold-warm exposure. The concept of their work is of broad interest and the data they present are appealing. However, at the current state, the study has some weak points that limit its significance and need to be further investigated.

Major comments:

1. What mechanisms do these authors propose for the interaction of RPS-2 and ZTF-7 interaction? It should be noted that ZTF-7 is located predominantly in the cytoplasma. The fact that RPS-2 could interact with ZTF-7 implicates that RPS-2 is also located in the cytoplasma.

2. Knockdown of rps-2 by RNAi induced a significant decrease in ZTF-7 expression. These authors should clarify how RPS-2 regulates the expression of ZTF-7. This regulation occurs at the transcriptional or post-transcriptional level? The authors need to determine the mRNA expression of ztf-7 in worms subjected to rps-2 RNAi. Alternatively, interaction of RPS-2 with ZTF-7 probably promotes ZTF-7 protein stability.

3. What is the physiological significance of cold-warm exposure-induced exosome translocation from the nucleolus to the nucleoplasm in this tissue? Does blockage of this exosome translocation influence life-history traits in worms.

4. The experiments in figure 2E and 2F lack important controls. It is hard to understand the importance of these results in figure 2. If the authors do not have additional comments that would highlight their significance, they should be transferred to a supplemental figure.

Minor issues:

1. Line 648, “C27F2.4” should be “C27F2.4::GFP”.

2. What type of tissue are these images taken from?

3. The quantification of data in figure 6D-F is lacking.

4. Which band corresponding to ZTF-7::GFP::3xFLAG in figure 6B and 6C?

5. There are many grammar errors in this manuscript.

Reviewer #2: In this highly interesting paper Xu el al. discover that cold-warm exposure translocates the exosome from the nucleolus to the nucleoplasm. They demonstrate this using a number of different and convincing experiments, and also have the proper controls to make sure this effect is specific to some nuceolus components but not others. This is especially interesting in light of their previous seminal work on risiRNAs and their links to ribosomal RNA processing and regulation. Then, using an unbiased genetic screen, they identify a conserved ZTF-7/RPS-2 complex which controls this cold-warm response. They validated the results of their screen and the target that they identified and proceed to biochemical analysis of the protein finding the relevant proteins with which it co-localizes and interacts. KD of these co-binding proteins cancels cold-warm controlled exosome translocation. In summary, using very well done experiments they find a novel and physiologically relevant cold response mechanism in animals. I don’t have any suggestions for additional experiments, in my opinion the paper is very interesting and would generate follow up studies in its current form.

Reviewer #3: This manuscript by Xu, et al. explores the response to transient cold exposure followed by return to the preferred growth temperature (cold-warm shock) in the well-studied genetic model system C. elegans. They report that the sub-nuclear localization of RNA exosome subunits changes in response to cold-warm shock. This shift requires the primarily cytoplasmic protein ZTF-7 and its interactor RPS-2, and may define a new response pathway.

Overall, this manuscript is very well-written and the science was rigorous and convincing – a pleasure to read! There are a few places where the story could benefit from some additional clarity:

1. Figure 1c and many other graphs show the “percentage of nucleolar localization” on the y axis. Does this mean percent of nuclei with any nucleolar localization or does it indicate what fraction of the protein is in the nucleolus for any given nucleus (on average)? Presumably the former, but please clarify this.

2. Figure 3b shows that for all three alleles of ztf-7 the exos-10 is not depleted from the nucleolus by cold-warm shock. The control nuclei at 20C show a visually apparent enrichment of exos-10 in the nucleolus. The ust119 nuclei appear to as well, but for ust117 and ust118, the fluorescence looks uniform. Is this just how these particular images appear, or is this a real difference? The line graphs of fluorescence intensity across the diameter of a nucleus as shown in Fig 1b, d, and f would be interesting to clarify this.

3. Figure 3h shows a normal lifespan of ztf-7 at 20C. Most published lifespans are performed at 25C – is the ztf-7 lifespan wild type at 25C as well?

4. Please clarify whether the nuclei in Figure 5a are from worms grown at 20C with no cold-warm shock.

5. Are all cold-warm shock images taken at the 0.5 hour recovery timepoint – e.g. Fig 5c? Please clarify in the figure legend or elsewhere.

6. When partial RNAi depletion is done, it is typical to measure mRNA levels by qPCR to assess how strong the knock down is. Can this be done for the partial RNAi experiments in this paper, e.g. Figure 5c?

7. Figure 6c is a little confusing since the IP-western is not labeled but doesn’t align exactly with the western of the input. Please either label both westerns or adjust the figure so that the blot slices are the same size and alignment.

8. What cell type are we looking at throughout this manuscript?

9. Please include statistical analyses, when appropriate. The methods section says a Student’s T-test or unpaired Wilcoxon test was done as indicated, but this is not indicated anywhere in the manuscript. Which data were analyzed statistically? Only Figure 3f and g? What test was done? This should be indicated in the figure legend. At minimum, figures 5c, 7b, 7c, 7d, and 7e should also have a statistical analysis.

**Have all data underlying the figures and results presented in the manuscript been provided?**

Reviewer #1: Yes

Reviewer #2: Yes

Reviewer #3: Yes

PLOS authors have the option to publish the peer review history of their article (what does this mean?). If published, this will include your full peer review and any attached files.

Reviewer #1: No

Reviewer #2: **Yes: **Oded Rechavi

Reviewer #3: No

---

## [Decision Letter · Decision Letter 1]

20 Jan 2023

Dear Dr guang,

We are pleased to inform you that your manuscript entitled "A ZTF-7/RPS-2 complex mediates the cold-warm response in C. elegans" has been editorially accepted for publication in PLOS Genetics. Congratulations!

Yours sincerely,

Dengke K Ma

Guest Editor

PLOS Genetics

Gregory P. Copenhaver

Editor-in-Chief

PLOS Genetics

Comments from the reviewers (if applicable):

Reviewer's Responses to Questions

**Comments to the Authors:**

Reviewer #1: The authors properly addressed my concerns. Well done!

Reviewer #2: I think this is an excellent paper. I didn't have any significant concern also when I read the original version, and I am also happy with the revised version. I hope the paper will be accepted without delay.

Reviewer #3: This revised manuscript addresses all previous suggestions by adding clarifying details and experiments that strengthen their conclusions. The authors convincingly describe RNA exosome dynamics in response to cold-warm shock. They identify a novel stress response pathway in which the exosome translocates from nucleoli to nucleoplasm. This is mediated by the Zn-finger protein ZTF-7 and several components of the ribosomal small subunit.

**Have all data underlying the figures and results presented in the manuscript been provided?**

Reviewer #1: None

Reviewer #2: Yes

Reviewer #3: Yes

PLOS authors have the option to publish the peer review history of their article (what does this mean?). If published, this will include your full peer review and any attached files.

Reviewer #1: No

Reviewer #2: **Yes: **Oded Rechavi

Reviewer #3: No

**Data Deposition**

http://datadryad.org/submit?journalID=pgenetics&manu=PGENETICS-D-22-01123R1

**Press Queries**

---

## [Editor Report · Acceptance letter]

6 Feb 2023

PGENETICS-D-22-01123R1 

A ZTF-7/RPS-2 complex mediates the cold-warm response in C. elegans 

Dear Dr Guang, 

We are pleased to inform you that your manuscript entitled "A ZTF-7/RPS-2 complex mediates the cold-warm response in C. elegans" has been formally accepted for publication in PLOS Genetics! Your manuscript is now with our production department and you will be notified of the publication date in due course.

With kind regards,

Bernadett Koltai

PLOS Genetics

On behalf of:
